# Hierarchical Clustering Beyond the Worst-Case

**Vincent Cohen-Addad**
University of Copenhagen
vcohenad@gmail.com

**Varun Kanade**
University of Oxford
Alan Turing Institute
varunk@cs.ox.ac.uk

**Frederik Mallmann-Trenn**
MIT
mallmann@mit.edu

## Abstract

Hiererachical clustering, that is computing a recursive partitioning of a dataset to obtain clusters at increasingly finer granularity is a fundamental problem in data analysis. Although hierarchical clustering has mostly been studied through procedures such as linkage algorithms, or top-down heuristics, rather than as optimization problems, Dasgupta [9] recently proposed an objective function for hierarchical clustering and initiated a line of work developing algorithms that explicitly optimize an objective (see also [7, 22, 8]). In this paper, we consider a fairly general random graph model for hierarchical clustering, called the hierarchical stochastic block model (HSBM), and show that in certain regimes the SVD approach of McSherry [18] combined with specific linkage methods results in a clustering that give an $O(1)$ approximation to Dasgupta's cost function. Finally, we report empirical evaluation on synthetic and real-world data showing that our proposed SVD-based method does indeed achieve a better cost than other widely-used heurstics and also results in a better classification accuracy when the underlying problem was that of multi-class classification.

## 1 Introduction

Computing a recursive partitioning of a dataset to obtain a finer and finer classification of the data is a classic problem in data analysis. Such a partitioning is often refered to as a *hierarchical clustering* and represented as a rooted tree whose leaves correspond to data elements and where each internal node induces a cluster of the leaves of its subtree. There exists a large literature on the design and analysis of algorithms for hierarchical clustering (see *e.g.,* [21]). Two main approaches have proven to be successful in practice so far: on the one hand *divisive* heuristics compute the hierarchical clustering tree in a top-down fashion by recursively partitioning the data (see *e.g.,* [14]). On the other hand, *agglomerative* heuristics produce a tree by first defining a cluster for each data elements and successively merging clusters according to a carefully defined function (see *e.g.,* [19]). These heuristics are widely used in practice and are now part of the data scientists' toolkit—standard machine learning libraries contain implementations of both types of heuristics.

Agglomerative heuristics have several appealing features: they are easy to implement, easy to tune, and their running time is $\widetilde{O}(n^2 \mathrm{poly}\log n)$ on a dataset of size $n$. Standard divisive heuristics based on graph partitioning or clustering methods (like for example the bisection $k$-means or the recursive sparsest-cut approaches) often involve solving or approximating NP-hard problems.[1] Therefore, it is natural to

ask how good the solution output by an agglomerative method is compared to the solution output by a top-down method.

From a qualitative perspective, this question has been addressed in a large body of work (see *e.g.,* [5]). However, from a quantitative perspective little is known. As Dasgupta observes in his recent work [9], both agglomerative and divisive heuristics are defined procedurally rather than in term of an objective function to optimize, a reason why a quantitative comparision of the different heuristics is rather difficult. Dasgupta introduced an objective function to model the problem of finding a hierarchical clustering of a similarity graph—such an objective can be used to explicitly design optimization algorithms that minimize this cost function as well as serve as a quantitative measure of the quality of the output.

Given a similarity graph *i.e.,* a graph where vertices represent data elements and edge weights similarities between data elements, Dasgupta's objective function associates a cost to any hierarchical clustering tree of the graph. He showed that his objective function exhibits several desirable properties: For example, if the graph is disconnected *i.e.,* data elements in different connected components are very dissimilar, a tree minimizing this objective function will first split the graph according to the connected components.

This axiomatic approach to defining a "meaningful" objective function for hierarchical clustering has been further explored in recent work by Cohen-Addad *et al.* [8]. Roughly speaking, they characterize a family of cost functions, which includes Dasgupta's cost function, that when the input graph has a "natural" ground-truth hierarchical clustering tree (in other words a natural classification of the data), this tree has optimal cost (and any tree that is not a "natural" hierarchical clustering tree of the graph has higher cost). Therefore, the results by Dasgupta and Cohen-Addad *et al.* indicate that Dasgupta's cost function provides a sound framework for a rigorous quantitative analysis of agglomerative and divisive heuristics.

A suitable objective function to measure the quality of a clustering also allows one to explicitly design algorithms that minimize the cost. Dasgupta showed that the recursive sparsest-cut heuristic is an $O(\log^{3/2} n)$-approximation algorithm for his objective function. His analysis has been improved by Charikar and Chatziafratis [7] and Cohen-Addad et al. [8] to $O(\sqrt{\log n})$. Unfortunately, Charikar and Chatziafratis [7] and Roy and Pokutta [22] showed that, for general inputs, the problem cannot be approximated within any constant factor under the Small-Set Expansion hypothesis. Thus, as suggested by Charikar and Chatziafratis [7], a natural way to obtain a more fine-grained analysis of the classic agglomerative and divisive heuristics is to study *beyond-worst* case scenarios.

**Random Graph Model for Hierarchical Clustering**. A natural way to analyse a problem beyond the worst-case is to consider a suitable random input model, which is the focus of this paper. More precisely, we introduce a random graph model based on the notion of "hierarchical stochastic block model" (HSBM) introduced by Cohen-Addad *et al.*, which is a natural extensions of the stochastic block model introduced. Our random graph model relies on the notion of *ultrametric*, a metric in which the triangle inequality is strengthened by requiring $d(x,y) \leqslant \max(d(x,z), d(y,z))$. This is a key concept as ultrametrics exactly capture the notion of data having a "natural" hierarchical structure (cf. [5]). The random graphs are generated from data that comes from an ultrametric, but the randomness hides the natural hierarchical structure. Two natural questions are: Given a random graph generated in such a fashion, when is it possible to identify the underlying ultrametric and is the optimization of Dasgupta's cost function easier for graphs generated according to such a model. The former question was partially addressed by Cohen-Addad *et al.* and our focus is primarily on developing algorithms that achieve an $O(1)$ approximation to the expected Dasgupta cost, not on recovering the underlying ultrametric.

More formally, assume that the data elements lie in an unknown ultrametric space $(A, \text{dist})$ and so exhibit a natural hierarchical clustering defined by this ultrametric. The input is a random graph generated as follows: an edge is added between nodes $u, v \in A$ with probability $p = f(\text{dist}(u,v))$, where $f$ is an (unknown) non-increasing function with range $(0,1)$. Thus, vertices that are very close in the ultrametric (and so very similar) have a higher probability to have an edge between them than vertices that are further apart. Given such a random graph, the goal is to obtain a hierarchical clustering tree that has a good cost for the objective function. The *actual* ground-truth tree is optimal in expectation and we focus on designing algorithms that with high probability output a tree whose cost is within a constant factor of the expected cost of the ground-truth tree. Although, we do not study it in this work, the question of exact recovery is also an interesting one and the work of Cohen-Addad *et al.* [8] addresses this partially in certain regimes.

**Algorithmic Results**. Even in the case of random graphs, the linkage algorithms may perform quite poorly, mainly because ties may be broken unfavourably at the very bottom, when the clusters are

singleton nodes; these choices cannot be easily compensated later on in the algorithm. We thus consider the LINKAGE++ algorithm which first uses a *seeding step* using a standard SVD approach to build clusters of a significant size, which is an extension of the algorithm introduced in [8]. Then, we show that using these clusters as starting point, the classic single-linkage approach achieves a $(1+\varepsilon)$-approximation for the problem (cf. Theorem 2.4).

**Experimental Results**. We evaluate the performance of LINKAGE++ on real-world data (Scikit-learn) as well as on synthetic hierarchical data. The measure of interest is the Dasgupta cost function and for completeness we also consider the classification error (see *e.g.,* [22]). Our experiments show that 1) LINKAGE++ performs well on all accounts and 2) that a clustering with a low Dasgupta cost appears to be correlated with a good classification. On synthetic data LINKAGE++ seems to be clearly superior.

**Related Work**. Our work follows the line of research initiated by Dasgupta [9] and further studied by [22, 7, 8]. Dasgupta [9] introduced the cost function studied in this paper and showed that the recursive sparsest-cut approach yields an $O(\log^{3/2} n)$. His analysis was recently improved to $O(\sqrt{\log n})$ by [7, 8]. Roy and Pokutta [22] and Charikar also considered LP and SDP formulations with spreading constraints to obtain approximation algorithms with approximation factor $O(\log n)$ and $O(\sqrt{\log n})$ respectively. Both these works also showed the infeasibility of constant factor approximations under the small-set expansion hypothesis. Cohen-Addad *et al.* [8] took an axiomatic approach to identify suitable cost functions for data generated from ultrametrics, which results in a natural ground-truth clustering. They also looked at a slightly less general hierarchical stochastic blockmodel (HSBM), where each bottom-level cluster must have a linear size and with stronger conditions on allowable probabilities. Their algorithm also has a "seeding phase" followed by an agglomerative approach. We go beyond their bounds by focusing on approximation algorithms (we obtain a $(1+\varepsilon)$-approximation) whereas they aim at recovering the underlying ultrametric. As the experiments show, this trade-off seem not to impact the classification error compared to classic other approaches.

There is also a vast literature on graph partitionning problems in random and semi-random models. Most of this work (see *e.g.,* [18, 11]) focuses on recovering a hidden subgraph *e.g.,* a clique, whereas we address the problem of obtaining good approximation guarantees w.r.t. an objective function.the reader may refer to [24, 13] for the definitions and the classic properties on agglomerative and divisive heuristics. Agglomerative and divisive heuristics have been widely studied from either a qualitative perspective or for classic "flat" clustering objective like the classic $k$-median and $k$-means, see *e.g.,* [20, 10, 16, 3, 2]. For further background on hierarchical clustering and its application in machine learning and data science, the reader may refer to *e.g.,* [15, 23, 12, 6].

**Preliminaries** In this paper, we work with undirected weighted graph $G = (V, E, w)$, where $V$ is a set of vertices, $E$ a set of edges, and $w : E \to \mathbb{R}_+$. In the random and semi-random model, we work with unweighted graphs. We slightly abuse notation and extend the function $w$ to subsets of $V$. Namely, for any $A, B \subseteq V$, let $w(A, B) = \sum_{a \in A, b \in B} w(a, b)$. We use weights to model similarity, namely $w(u, v) > w(u, w)$ means that data element $u$ is more similar to $v$ than to $w$. When $G$ is clear from the context, we let $|V| = n$ and $|E| = m$. For any subset $S$ of vertices of a graph $G$, let $G[S]$ be the subgraph induced by the nodes of $S$.

In the following, let $G = (V, E, w)$ be a weighted graph on $n$ vertices. A *cluster tree* or *hierarchical clustering* $T$ for $G$ is a rooted binary tree with exactly $|V|$ leaves, each of which is labeled by a distinct vertex $v \in V$. We denote $\text{LCA}_T(u, v)$ the lowest common ancestor of vertices $u, v$ in $T$. Given a tree $T$ and a node $N$ of $T$, we say that the subtree of $N$ in $T$ is the the connected subgraph containing all the leaves of $T$ that are descendant of $N$ and denote this set of leaves by $V(N)$. A metric space $(X, d)$ is an ultrametric if for every $x, y, z \in X$, $d(x, y) \leqslant \max\{d(x, z), d(y, z)\}$.

We borrow the notion of a (similarity) *graph generated from an ultrametric* and *generating tree* introduced by [8]. A weighted graph $G = (V, E, w)$ is a generated from an ultrametric, if there exists an ultrametric $(X, d)$, such that $V \subseteq X$, and for every $x, y \in V, x \neq y$, $e = \{x, y\}$ exists, and $w(e) = f(d(x, y))$, where $f : \mathbb{R}_+ \to \mathbb{R}_+$ is a non-increasing function.

**Definition 1.1** (Generating Tree). *Let $G = (V, E, w)$ be a graph generated by a minimal ultrametric $(V, d)$. Let $T$ be a rooted binary tree with $|V|$ leaves; let $\mathcal{N}$ denote the internal nodes and $L$ the set of leaves of $T$ and let $\sigma : L \to V$ denote a bijection between the leaves of $T$ and nodes of $V$. We say that $T$ is a generating tree for $G$, if there exists a weight function $W : \mathcal{N} \to \mathbb{R}_+$, such that for $N_1, N_2 \in \mathcal{N}$, if $N_1$ appears on the path from $N_2$ to the root, $W(N_1) \leqslant W(N_2)$. Moreover for every $x, y \in V$, $w(\{x, y\}) = W(LCA_T(\sigma^{-1}(x), \sigma^{-1}(y)))$.*

As noted in [8], the above notion bear similarities to what is referred to as a *dendrogram* in the machine learning literature (see *e.g.,* [5]).

**Objective Function**. We consider the objective function introduced by Dasgupta [9]. Let $G = (V, E, w)$ be a weighted graph and $T = (\mathcal{N}, \mathcal{E})$ be any rooted binary tree with leaves set $V$. The cost induced by a node $N$ of $T$ is $\text{cost}_T(N) = |V(N)| \cdot w(V(C_1), V(C_2))$ where $C_1, C_2$ are the children of $N$ in $T$. The cost of $T$ is $\text{cost}_T = \sum_{N \in \mathcal{N}} \text{cost}_T(N)$. As pointed out by Dasgupta [9], this can be rephrased as $\text{cost}_T = \sum_{(u,v) \in E} w(u,v) \cdot |V(\text{LCA}_T(u,v))|$.

## 2 A General Hierarchical Stochastic Block Model

We introduce a generalization of the HSBM studied by [8] and [17]. Cohen-Addad et al. [8] introduce an algorithm to recover a "ground-truth" hierarchical clustering in the HSBM setting. The regime in which their algorithm works is the following: (1) there is a set of hidden clusters that have linear size and (2) the ratio between the minimum edge probability and the maximum edge probability is $O(1)$. We aim at obtaining an algorithm that "works" in a more general setting. We reach this goal by proposing on $(1 + \varepsilon)$-approximation algorithms. Our algorithm very similar to the widely-used linkage approach and remains easy to implement and parallelize. Thus, the main message of our work is that, on "structured inputs" the agglomerative heuristics perform well, hence making a step toward explaining their success in practice.

The graphs generated from our model possess an underlying, hidden (because of noise) "ground-truth hierarchical clustering tree" (see Definition 2.1). This aims at modeling real-world classification problem for which we believe there is a natural hierarchical clustering but perturbed because of missing information or measurement erros. For example, in the tree of life, there is a natural hierarchical clustering hidden that we would like to reconstruct. Unfortunately because of extinct species, we don't have a perfect input and must account for noise. We formalize this intuition using the notion of generating tree (Def 1.1) which, as hinted at by the definition, can be associated to an ultrametric (and so a "natural" hierarchical clustering). The "ground-truth tree" is the tree obtained from a generating tree on $k$ leaves to which we will refer as "bottom"-level clusters containing $n_1, n_2, ..., n_k$ nodes (following the terminology in [8]). Each edge of a generated graph has a fixed probability of being present, which only depends on the underlying ground-truth tree. This probability is a function of the clusters in which their endpoints lie and the underlying graph on $k$ vertices for which the generating tree is generating (as in Def 1.1).

**Definition 2.1** (Hierarchical Stochastic Block Model – Generalization of [8]). *Let $n$ be a positive integer. A hierarchical stochastic block model with $k$ bottom-level clusters is defined as follows:*

*1) Let $\widetilde{G}_k = (\widetilde{V}_k, \widetilde{E}_k, w)$ be a graph generated from an ultrametric, where $|\widetilde{V}_k| = k$ for each $e \in \widetilde{E}_k$, $w(e) \in (0,1)$. let $\widetilde{T}_k$ be a tree on $k$ leaves, let $\widetilde{\mathcal{N}}$ denote the internal nodes of $\widetilde{T}$ and $\widetilde{L}$ denote the leaves; let $\widetilde{\sigma} : \widetilde{L} \to [k]$ be a bijection. Let $\widetilde{T}$ be generating for $\widetilde{G}_k$ with weight function $\widetilde{W} : \widetilde{\mathcal{N}} \to [0,1)$.*

*2) For each $i \in [k]$, let $p_i \in (0,1]$ be such that $p_i > \widetilde{W}(N)$, if $N$ denotes the parent of $\widetilde{\sigma}^{-1}(i)$ in $\widetilde{T}$.*

*3) For each $i \in [k]$, there is a positive integer $n_i$ such that $\sum_{i=1}^{k} n_i = n$.*

*Then a random graph $G = (V, E)$ on $n$ nodes is defined as follows. Each vertex $i \in [n]$ is assigned a label $\psi(i) \in [k]$, so that exactly $n_j$ nodes are assigned the label $j$ for $j \in [k]$. An edge $(i, j)$ is added to the graph with probability $p_{\psi(i)}$ if $\psi(i) = \psi(j)$ and with probability $\widetilde{W}(N)$ if $\psi(i) \neq \psi(j)$ and $N$ is the least common ancestor of $\widetilde{\sigma}^{-1}(i)$ and $\widetilde{\sigma}^{-1}(j)$ in $\widetilde{T}$. The graph $G = (V, E)$ is returned without any labels.*

We use, for a generating tree $\widetilde{T}$, the notation $p_{\min}$ to denote $\widetilde{W}(N_0)$, where $N_0$ is the root node of $\widetilde{T}$. Let $n_{\min}$ be the size of the smallest cluster (of the $k$ clusters) As in [8], we will use the notion of *expected graph*. The *expected graph* as the is the weighted complete graph $\bar{G}$ in which an edge $(i, j)$ has weight $p_{i,j}$, where $p_{i,j}$ is the probability with which it appears in the random graph $G$. We refer to any tree that is generating for the expected graph $\bar{G}$ as a *ground-truth tree* for $G$. In order to avoid ambiguity, we denote by $\text{cost}_T(G)$ and $\text{cost}_T(\bar{G})$ the costs of the cluster tree $T$ for the unweighted (random) graph $G$ and weighted graph $\bar{G}$ respectively. Observe that due to linearity of expectation for any tree $T$ and any admissible cost function, $\text{cost}_T(\bar{G}) = \mathbb{E}[\text{cost}_T(G)]$, where the expectation is with respect to the random choices of edges in $G$. We have

**Theorem 2.2.** *Let $n$ be a positive integer and $p_{\min} = \omega(\sqrt{\log n / n})$. Let $k$ be a fixed constant and $G$ be a graph generated from an HSBM (as per Defn. 2.1) where the underlying graph $\widetilde{G}_k$ has $k$ nodes and minimum probability is $p_{\min}$. For any binary tree $T$ with $n$ leaves labelled by the vertices of $G$,*

*the following holds with high probability:* $|cost(T) - \mathbb{E}[cost(T)]| \leqslant o(\mathbb{E}[cost(T)])$. *The expectation is taken only over the random choice of edges. In particular if $T^*$ is a ground-truth tree for $G$, then, with high probability,* $cost(T^*) \leqslant (1+o(1)) \min_{T'} cost(T') = (1+o(1)) OPT$.

**Algorithm LINKAGE++, a $(1+\varepsilon)$-Approximation Algorithm in the HSBM**. We consider a simple algorithm, called LINKAGE++, which works in two phases (see Alg. 1). We use a result of [18] who considers the planted partition model. His approach however does not allow to recover directly a hierarchical structure when the input has it.

---

**Algorithm 1** LINKAGE++

1: **Input:** Graph $G = (V,E)$ generated from an HSBM.
2: **Parameter:** An integer $k$.
3: Apply (SVD) projection algorithm of [18, Thm. 12] with parameters $G$, $k$, $\delta = |V|^{-2}$, to get $\zeta(1),...,\zeta(|V|) \in \mathbb{R}^{|V|}$ for vertices in $V$, where $\dim(\text{span}(\zeta(1),...,\zeta(|V|))) = k$.
4: Run the single-linkage algorithm on the points $\{\zeta(1),...,\zeta(|V|)\}$ until there are exactly $k$ clusters. Let $\mathcal{C} = \{C_1^\zeta,...,C_k^\zeta\}$ be the clusters (of points $\zeta(i)$) obtained. Let $C_i \subseteq V$ denote the set of vertices corresponding to the cluster $C_i^\zeta$.
5: Define $\text{dist}: \mathcal{C} \times \mathcal{C} \mapsto \mathbb{R}_+$: $\text{dist}(C_i^\zeta, C_j^\zeta) = w(C_i^\zeta, C_j^\zeta)/(|C_i^\zeta||C_j^\zeta|)$.
6: **while** there are at least two clusters in $\mathcal{C}$ **do**
7: $\quad$ Take the pair of clusters $C_i', C_j'$ of $\mathcal{C}$ at max $\text{dist}(C_i', C_j')$. Define a new cluster $C' = \{C_i' \cup C_j'\}$.
8: $\quad$ Update dist: $\text{dist}(C', C_\ell') = \max(\text{dist}(C_i', C_\ell'), \text{dist}(C_j', C_\ell'))$
9: $\quad$ $\mathcal{C} \leftarrow \mathcal{C} \setminus \{C_i'\} \setminus \{C_j'\} \cup \{C'\}$
10: **end while**
11: The sequence of merges in the while-loop (Steps 6 to 10) induces a hierarchical clustering tree on $\{C_1^\zeta,...,C_k^\zeta\}$, say $T_k'$ with $k$ leaves $(C_1^\zeta,...,C_k^\zeta)$. Replace each leaf $C_i^\zeta$ of $T_k'$ by the tree obtained for $C_i^\zeta$ at Step 4 to obtain $T$.
12: Repeat the algorithm $k' = 2k \log n$ times. Let $T^1,...T^{k'}$ be the corresponding outputs.
13: **Output:** Tree $T^i$ (out of the $k'$ candidates) that minimises $\Gamma(T_i)$.

---

**Theorem 2.3** ([18], Observation 11 and a simplification of Theorem 12). *Let $\delta$ be the confidence parameter. Assume that for all $u,v$ belonging to different clusters with adjacency vectors $\mathbf{u},\mathbf{v}$ (i.e., $u_i$ is 1 if the edge $(u,i)$ exists in $G$ and $0$ otherwise) satisfy*

$$\|\mathbb{E}[\mathbf{u}] - \mathbb{E}[\mathbf{v}]\|_2^2 \geqslant c \cdot k \cdot \left(\sigma^2 n / n_{\min} + \log(n/\delta)\right) \tag{1}$$

*for a large enough constant $c$, where $\mathbb{E}[\mathbf{u}]$ is the entry-wise expectation and $\sigma^2 = \omega(\log^6 n/n)$ is an upper bound on the variance. Then, the algorithm of [18, Thm. 12] with parameters $G,k,\delta$ projects the columns of the adjacency matrix of $G$ to points $\{\zeta(1),...,\zeta(|V|)\}$ in a $k$-dimensional subspace of $\mathbb{R}^{|V|}$ such that the following holds w.p. at least $1-\delta$ over the random graph $G$ and with probability $1/k$ over the random bits of the algorithm. There exists $\eta > 0$ such that for any $u$ in the $i$th cluster and $v$ in the $j$th cluster: 1) if $i = j$ then $\|\zeta(u) - \zeta(v)\|_2^2 \leqslant \eta$ and 2) if $i \neq j$ then $\|\zeta(u) - \zeta(v)\|_2^2 > 2\eta$.*

In the remainder we assume $\delta = 1/|V|^2$. We are ready to state our main theorem.

**Theorem 2.4.** *Let $n$ be a positive integer and $\varepsilon > 0$ a constant. Assume that the separation of bottom clusters given by (1) holds, $p_{\min} = \omega(\sqrt{\log n/n})$, and $n_{\min} \geqslant \sqrt{n} \cdot \log^{1/4} n$. Let $k$ be a fixed constant and $G$ be a graph generated from an HSBM (as per Defn. 2.1) where the underlying graph $\widetilde{G}_k$ has $k$ nodes with satisfying the above constraints.*

*With high probability, Algorithm 1 with parameter $k$ on graph $G$ outputs a tree $T'$ that satisfies $cost_{T'} \leqslant (1+\varepsilon) OPT$.*

We note that $k$ might not be known in advance. However, different values of $k$ can be tested and an $O(1)$-estimate on $k$ is enough for the proofs to hold. Thus, it is possible to run Algorithm 1 $O(\log n)$ times with different "guesses" for $k$ and take the best of these runs.

Let $G = (V,E)$ be the input graph generated according to an HSBM. Let $T$ be the tree output by Algorithm 1. We divide the proof into two main lemmas that correspond to the outcome of the two phases mentioned above.

The algorithm of [18, Thm. 12] might fail for two reasons: The first reason is that the random choices by the algorithm result in an incorrect clustering. This happens w.p. at most $1-1/k$ and we can simply repeat the algorithm sufficiently many times to be sure that at least once we get the desired result, *i.e.*, the projections satisfy the conclusion of Thm. 2.3. Lemmas 2.6, 2.7 show that in this case, Steps 6 to 10 of LINKAGE++ produce a tree that has cost close to optimal. Ultimately, the algorithm simply outputs a tree that has the least cost among all the ones produced (and one of them is guaranteed to have cost $(1+\varepsilon)$OPT) with high probability.

The second reason why the McSherry's algorithm may fail is that the generated random graph $G$ might "deviate" too much from its expectation. This is controlled by the parameter $\delta$ (which we set to $1/|V|^2$). Deviations from expected behaviour will cause our algorithm to fail as well. We bound this failure probability in terms of two events. The first bad event is that McSherry's algorithm fails for either of the aforementioned reasons. We denote the complement of this event $\mathcal{E}_1$. The second bad event it that the number of edges between the vertices of two nodes of the ground-truth tree deviates from it's expectation. Namely, that given two nodes $N_1, N_2$ of $T^*$, we expect the cut to be $E_{(N_1,N_2)} = |V(N_1)| \cdot |V(N_1)| \cdot W(\text{LCA}_{T*}(N_1,N_2))$. Thus, we define $\mathcal{E}_2$ to be the event that $|w(V(N_1),V(N_2)) - E_{(N_1,N_2)}| < \varepsilon^2 E_{(N_1,N_2)}$ for all cuts of the $k$ bottom leaves. Note that the number of cuts is bounded by $2^k$ and we will show that, due to size of $n_{\min}$ and $p_{\min}$ this even holds w.h.p.. The assumptions on the ground-truth tree will ensure that the latter holds w.h.p. allowing us to argue that both events hold w.p. at least $\Omega(1/k)$ Thus, from now on we assume that both "good" events $\mathcal{E}_1$ and $\mathcal{E}_2$ occur. We bound the probability of event $\mathcal{E}_1$ in Lemma 2.5. We now prove a structural properties of the tree output by the algorithm, we introduce the following definition. We say that a tree $T = (\mathcal{N}, \mathcal{E})$ is a *$\gamma$-approximate ground-truth tree* for $G$ and $T^*$ if there exists a weight function $W' : \mathcal{N} \mapsto \mathbb{R}_+$ such that for any two vertices $a, b$, we have that

1. $\gamma^{-1} W'(\text{LCA}_T(a,b)) \leqslant W(\text{LCA}_{T*}(a,b)) \leqslant \gamma W'(\text{LCA}_T(a,b))$ and

2. for any node $N$ of $T$ and any node $N'$ descendant of $N$ in $T$, $W(N) \leqslant W(N')$.

**Lemma 2.5.** *Let $G$ be generated by an* HSBM. *Assume that the separation of bottom clusters given by* (1) *holds. Let $C_1^*, ..., C_k^*$ be the hidden bottom-level clusters, i.e., $C_i^* = \{v \mid \psi(v) = i\}$. With probability at least $\Omega(1/k)$, the clusters obtained after Step 4 correspond to the assignment $\psi$, i.e., there exists a permutation $\pi : [k] \to [k]$, such that $C_j = C_{\pi(j)}^*$.*

**Lemma 2.6.** *Assume that the separation of bottom clusters given by* (1) *holds, $p_{\min} = \omega(\sqrt{\log n / n})$, and $n_{\min} \geqslant \sqrt{n} \cdot \log^{1/4} n$. Let $G$ be generated according to an* HSBM *and let $T^*$ be a ground-truth tree for $G$. Assume that events $\mathcal{E}_1$ and $\mathcal{E}_2$ occur, and that furthermore, the clusters obtained after Step 4 correspond to the assignment $\psi$, i.e., there exists a permutation $\pi : [k] \to [k]$ such that for each $v \in C_i$, $\psi(v) = \pi(i)$. Then, the output by the algorithm is a $(1+\varepsilon)$-approximate ground-truth tree.*

The following lemma allows us to bound the cost of an approximate ground-truth tree.

**Lemma 2.7.** *Let $G$ be a graph generated according to an* HSBM *and let $T^*$ be a ground-truth tree for $G$. Let $\bar{G}$ be the expected graph associated to $T^*$ and $G$. Let $T$ be a $\gamma$-approximate ground-truth tree. Then, $\text{cost}_T \leqslant \gamma^2 \text{OPT}$.*

*Proof of Theorem 2.4.* Conditioning on $\mathcal{E}_1$ and $\mathcal{E}_2$ which occur w.h.p. and combining Lemmas 2.5, 2.7, and 2.6 together with Theorem 2.2 yields the result. As argued before, $\mathcal{E}_1$ holds at least w.p. $1/k$ and it is possible to boost part of this probability by running Algorithm 1 multiple times. Running it $\Omega(k \log n)$ times and taking the tree with the smallest cost yields the result. Moreover, $\mathcal{E}_2$ also holds w.h.p.. $\square$

## 3 Empirical Evaluation

In this section, we evaluate the effectiveness of LINKAGE++ on real-world and synthetic datasets. We compare our results to the classic agglomerative heuristics for hierarchical clustering both in terms of the cost function and the classification error. Our goal is answering the question: *How good is* LINKAGE++ *compared to the classic agglomerative approaches on real-world and synthetic data that exhibit a ground-truth clustering?*

**Datasets**. The datasets we use are part of the standard Scikit-learn library [4] (and most of them are available at the UCI machine learning repository [1]). Most of these datasets exhibit a "flat" clustering structure, with the exception of the `newsgroup` datasets which is truly hierarchical. The goal of the

algorithm is to perform a clustering of the data by finding the underlying classes. The datasets are: `iris`, `digits`, `newsgroup`[2], `diabetes`, `cancer`, `boston`. For a given dataset, we define similarity between data elements using the *cosine similarity*, this is a standard approach for defining similarity between data elements (see, *e.g.,* [22]) This induces a weighted similarity graph that is given as input to LINKAGE++.

**Synthethic Data**. We generate random graphs of sizes $n \in \{256, 512, 1024\}$ according to the model described in Section 2.1. More precisely, we define a binary tree on $\ell \in \{4,8\}$ bottom clusters/leaves. Each leaf represents a "class". We create $n/\ell$ vertices for each class. The probability of having an edge between two vertices of class $a$ and $b$ is given by the probability induced by lowest common ancestor between the leaves corresponding to $a$ and $b$ respectively. We first define $p_{\min} = 2\log n \cdot \ell/n$. The probability induced by the vertices of the binary tree are the following: the probability at the root is $p = p_{\min} + (1 - p_{\min})/\log(\ell)$, and the probability induced by a node at distance $d$ from the root is $(d+1)p$. In particular, the probability induced by the leaves is $p_{\min} + \log(\ell)(1 - p_{\min})/\log(\ell) = 1$. We also investigate a less structured setting using a ground truth tree on three nodes.

**Method**. We run LINKAGE++ with 9 different breakpoints at which we switch between phase 1 and phase 2 (which corresponds to "guesses" of $k$). We output the clustering with the smallest cost. To evaluate our algorithm, we compare its performances to classic agglomerative heuristics (for the similarity setting): single linkage, complete linkage, (see also [24, 13] for a complete description) and to the approach of performing only phase 1 of LINKAGE++ until only one cluster remains; we will denote the approach as PCA+. Additionally, we compare ourselves to applying only phase 2 of LINKAGE++, we call this approach density-based linkage. We observe that the running times of the algorithms are of order $\widetilde{O}(n^2)$ stemming already from the agglomerative parts.[3] This is close to the $\widetilde{O}(n^2))$ running time achieved by the classic agglomerative heuristics.

We compare the results by using both the cost of the output tree w.r.t. the hierarchical clustering cost function and the *classification error*. The classification error is a classic tool to compare different (usually flat) clusterings (see, *e.g.,* [22]). For a $k$-clustering $C : V \mapsto \{1, ..., k\}$, the classification error w.r.t. a ground-truth flat clustering $C^* : V \mapsto \{1, ..., k\}$ is defined as $\min_{\sigma \in S_k} \left( \sum_{x \in V} \mathbf{1}_{C(x) \neq \sigma(C^*(x))} \right)/|V|$, where $S_k$ is the set of all permutations $\sigma$ over $k$ elements.

We note that the cost function is more relevant for the `newsgroup` dataset since it exhibits a truly hierarchical structure and so the cost function is presumably capturing the quality of the classification at different levels. On the other hand, the classification error is more relevant for the others data sets as they are intrinsically flat. All experiments are repeated at least 10 times and standard deviation is shown.

**Results**. The results are summarized in Figure 1, 2, and 3 (App. 3). Almost in all experiments LINKAGE++ performs extremely well w.r.t. the cost and classification error. Moreover, we observe that a low cost function correlates with a good classification error. For synthetic data, in both LINKAGE++ and PCA+, we observe in Figure 2b that classification error drops drastically from $k = 4$ to $k = 8$, from 0.5 to 0 as the size is number of nodes is increased from $n = 512$ to $n = 1024$. We observe this threshold phenomena for all fixed $k$ we considered. We can observe that the normalized cost in Figure 2a for the other linkage algorithms increases in the aforementioned setting.

Moreover, the only dataset where LINKAGE++ and PCA+ differ significantly is the hierarchical dataset `newsgroup`. Here the cost function of PCA+ is much higher. While the classification error of all algorithm is large, it turns out by inspecting the final clustering of LINKAGE++ and PCA+ that the categories which were being misclassified are mostly sub categories of the same category. On the dataset of Figure 3 (App. 3) only LINKAGE++ performs well.

**Conclusion**. Overall both algorithms LINKAGE++ and Single-linkage perform considerably better when it comes to real-world data and LINKAGE++ and PCA+ dominate on our synthetic datasets. However, in general there is no reason to believe that PCA+ would perform well in clustering truly hierarchical data: there are regimes of the HSBM for which applying only phase 1 of the algorithm might lead to a high missclassification error and high cost and for which we can prove that LINKAGE++ is an $(1+\varepsilon)$-approximation.

This is exemplified in Figure 3 (App. 3). Moreover, our experiments suggest that one should use in addition to LINKAGE++ other linkage algorithm and pick the algorithm with the lowest cost function, which appears to correlate with the classification error. Nevertheless, a high classification error of hierarchical

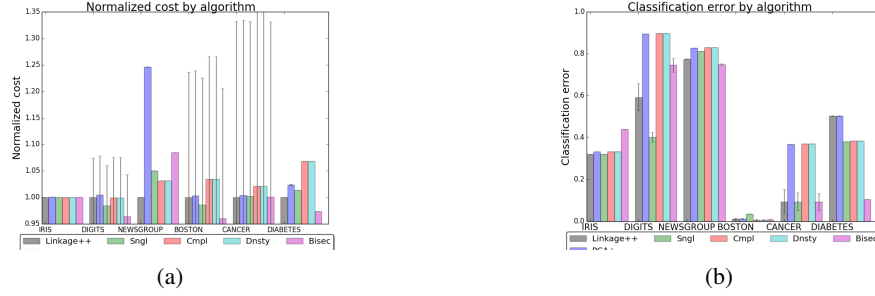

(a)                                                        (b)

Figure 1: A comparison of the algorithms on real-world data. (a) The figure shows the cost cost(·) of the algorithm normalized by the the cost of LINKAGE++. (b) The figure shows the percentage of misclassified nodes. By looking more closely at the output of the algorithm, one can see that a large fraction of the misclassifications happen in subgroups of the same group.

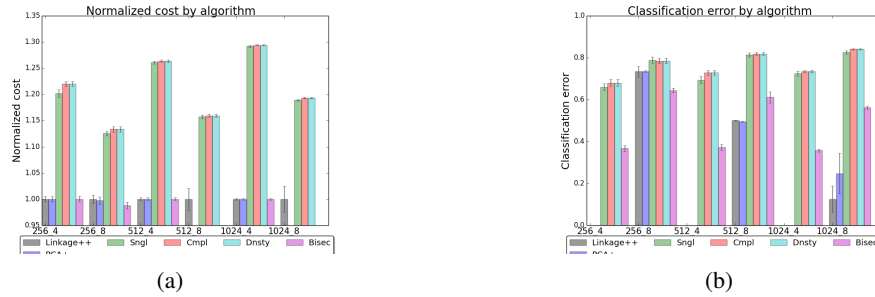

(a)                                                        (b)

Figure 2: A comparison of the algorithms on synthetic data for highly structured ground-truth for different $n,k$. PCA+ performs well on these inputs and we conjecture that this due to the highly structured nature of the ground-truth. (a) The cost of LINKAGE++ and PCA+ are well-below the costs' of the standard linkage algorithms. (b) We see a threshold phenomena for $k = 8$ from $n = 512$ to $n = 1024$. Here the classification error drops from $0.5$ to $0$, which is explained by concentration of the eigenvalues allowing the PCA to separated the bottom clusters correctly.

data is not a bad sign per se: A misclassification of subcategories of the same categories (as we observe in our experiments) is arguably tolerable, but ignored by the classification error. On the other hand, the cost function captures such errors nicely by its inherently hierarchical nature and we thus strongly advocate it.

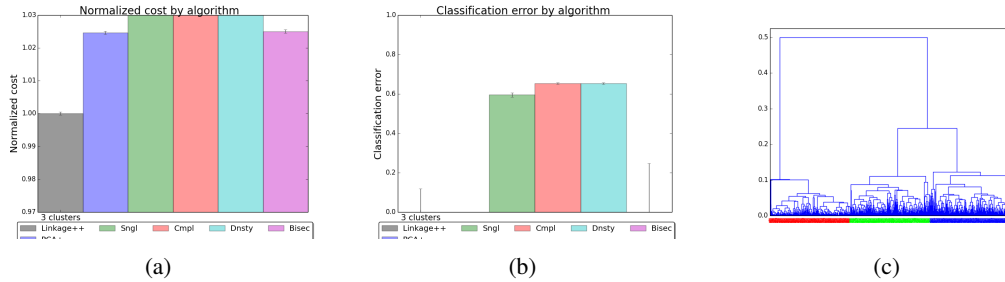

(a)                                    (b)                                    (c)

Figure 3: The clustering obtained by PCA+ on a ground truth tree on three nodes induced by the adjacency matrix $[[1.,0.49,0.39][0.49,0.49,0.39][0.39,0.39,0.62]]$ and $n = 999$ nodes split equally. Here only LINKAGE++ and PCA+ classify the bottom clusters of the subtrees correctly. However, the projection to the euclidian space (PCA) does not preserve the underlying ultramtric causing PCA+ to merge incorrectly. (a) LINKAGE++ recovers the ground truth. All other algorithm merge incorrectly. (b) LINKAGE++ and PCA+ classify the bottom clusters correctly causing the classification to be perfect even though PCA+ failed to correctly reconstruct the ground-truth. This suggests that the classification error is less suitable measure for hierarchical data. (c) PCA+ in contrast to LINKAGE++ merges incorrectly two bottom clusters of different branches in the ground-truth tree (green and blue as opposed to green and red).

**Acknowledgement**   The project leading to this application has received funding from the European Union's Horizon 2020 research and innovation programme under the Marie Sklodowska-Curie grant agreement No. 748094. This work was supported in part by EPSRC grant EP/N510129/1. This work was supported in part by NSF Award Numbers BIO-1455983, CCF-1461559, and CCF-0939370.

## Footnotes

[1]In some cases, it may be possible to have a very fast algorithms based on heuristics to compute partitions, however, we are unaware of any such methods that would have provable guarantees for the kinds of graphs that appear in hierarchical clustering.

[2] Due to the enormous size of the dataset, we consider a subset consisting of 'comp.graphics', 'comp.os.ms-windows.misc', 'comp.sys.ibm.pc.hardware', 'comp.sys.mac.hardware', 'rec.sport.baseball', 'rec.sport.hockey'

[3] Top $k$ singular vectors of an $n \times n$ matrix can be approximately computed in time $\widetilde{O}(kn^2)$.

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
