[Reviews · NeurIPS 2017]

Reviewer 1



This paper studies the problem of hierarchical clustering in a beyond-the-worst-case setting, where the data is a random graph generated by a Hierarchical Stochastic Block Model (HSBM). It proposes an SVD+linkage algorithm and proves that in certain regimes it returns a solution that is a constant approximation to the cost function of hierarchical clustering proposed by Dasgupta. It also considers an algorithm that combines the SDP relaxation approach and recursive sparsest cut approach in previous works to get a constant approximation in larger regimes. Roughly speaking, the HSBM model considered assumes some underlying tree T* over k leaves (each leaf corresponding to a bottom cluster containing a certain number of vertices). Each node in the tree has some weight in (0,1) and a node's weight should not be larger than its descendant's. Then generate a graph on the vertices where an edge (i,j) is added with probability equal to the weight of their lowest common ancestor (for those inside the bottom cluster, use the weight of the leaf). The algorithm is quite simple: first use SVD to get spectral embeddings of the vertices, and use single linkage on the Euclidean distances between the embeddings to get k clusters (aiming to recover the bottom clusters); then use single linkage to merge the k clusters but using the edge weights. The analysis uses the guarantee of the spectral embeddings and the assumptions that the bottom clusters are large enough to show that the bottom clusters are recovered. Then the algorithm will output a tree roughly corresponding to the underlying tree T*, which will have cost close to that of T* which is optimal. The result for the second algorithm follows from the existing results for the SDP relaxations in the random and semi-random settings, and those for the recursive sparsest cut approach to building the tree. The authors also provided an empirical evaluation on synthetic and real-world data. The results provide quite positive support for using the algorithm. The presentation is clear, though the notations are a bit heavy. Related works are properly cited and discussed. Overall it provides nice algorithms and analysis for hierarchical clustering assuming the data is from a reasonable random model. One of the algorithms is actually quite intuitive and has good empirical performance, thus provides a probable explanation for why simple hierarchical clustering heuristics work well in practice. minor: -- Line 172: very similar -> is very similar -- Line 191: \tilde{T}_k should be \tilde{T}? -- Line 230: delete "with"

Reviewer 2



Summary of paper: The paper studies hierarchical clustering in the recent framework of [1], which provides a formal cost function for the problem. [4] proposed a hierarchical stochastic block model (HSBM) for this problem, and showed that under this model, one can obtain a constant factor approximation of Dasgupta’s cost function [1]. The present work generalises the HSBM model slightly, where they do not restrict the number of clusters to be fixed, and hence, the cluster sizes may be sub-linear in the data size. The authors prove guarantees of a proposed algorithm under this model. They further extend their analysis to a semi-random setting where an adversary may remove some inter-cluster edges. They also perform experiments on some benchmark datasets to show the performance of the method. Quality: The paper is of decent quality. While the paper is theoretically sound, the experiment section is not very convincing. Why would one consider 5 flat cluster datasets and only one hierarchical dataset in a hierarchical clustering paper? Even for the last case, only a subset of 6 classes are used. Further, 3 out of 5 methods in comparison are variations of Linkage++, the other two being two standard agglomerative techniques. What about divisive algorithms? Clarity: Most parts of the paper can be followed, but in some places, the authors compressed the material a lot and one needs to refer to cited papers for intuitive understanding (for instance, Definition 2.1). There are minor typos throughout the paper, example in lines 41, 114, 126, 179, 201, 342. Originality: The paper derives a considerable portion from [4]. Though some extensions are claimed, I wonder if there is any technical challenges in proving the additional results (I did not see the details in supplementary). For example, is there any difference between Linkage++ and the method in [4]? Apparently I did not find any significant difference. The analysis on HSBM may allow growing k, but is there truly a challenge in doing so. At least for first step of algorithm, achieving n_min > \sqrt{n} does not seem a big hurdle. Analysis on the semi-random setting is new, but I wonder if there are any challenges in this scenario. Or does it immediately follow from combining HSBM analysis with standard techniques of semi-random graph partitioning? Significance: In general, hierarchical clustering in the framework of [1] has considerable significance, and [4] is also a quite interesting paper. However, I consider the present paper a minor extension of [4], with the only possible interesting aspect being the result in the semi-random setting. --------------------------- Update after rebuttal: I thank the authors for clarifying the technical challenges with proving the new results. Hence, I have now increased my score (though I am still concerned about the algorithmic and experimental contributions). Regarding the experiments, I do agree that there are not many well-defined benchmark out there. The papers that I know are also from early 2000s, but the authors may want to check the datasets used here: 1. Zhao, Karypis. Hierarchical Clustering Algorithms for Document Datasets. Data Mining and Knowledge Discovery, 2005 2. Alizadeh et al. Distinct types of diffuse large B-cell lymphoma identified by gene expression profiling. Nature, 2000 Apart from these, UCI repository has Glass and Reuters that I know have been used for hierarchical clustering. Perhaps, there are also more recent document / gene expression datasets in recent data mining papers.

Reviewer 3



This paper studies the hierarchical clustering cost function recently introduced by Dasgupta in a specific setting of both a general Hierarchical Stochastic Block Model (HSBM) and a semi-random extension. To achieve this they introduce an algorithm Linkage++ where first a single-linkage of an SVD-projection of the data is used for a first partitioning, and then these are clustered by single-linkage using edge-weights of the original data. Guarantees on the almost optimal cost of the result are proved. Empirical evaluations are performed both on the synthetic HSBM as well as on real-world data. The paper answers an important question of how well the cost can be approximated in a natural setting by a more-or-less standard algorithm. The benchmarks show that small costs can be achieved also for real-world data sets. I appreciate that the authors conclude that one should consider an ensemble of different algorithms to obtain decent approximations of the cost. The paper is written in a nice, clean, and understandable way. (As an observation: it is remarkable that single-linkage yields uniformly the best classification error for the real-world data sets in these experiments.) Questions --------- * For the synthetic data sets: can the true optimal cost be estimated? * Were the real-world data sets centered before calculating the cosine similarity? * How many singular values were considered? * Do I understand correctly that the difference in the experiments between "single-linkage" and "density-based linkage" algorithms is that the former uses say euclidean distance whereas the latter uses the cosine similarity? * Can you say something concretely on the run times of your experiments? Minor points ------------ * Definition 1.1: To me it is not clear what is meant by "MINIMAL ultrametric". If it is important it should be described quickly. * lines 318-321: log() seems to mean the binary logarithm, maybe write log_2() ? * line 201: "The expected graph _as the_ is the..." delete "as the"